# Observation of cavity-tunable topological phases of polaritons

Dong Zhao[1,7], Ziyao Wang[1,7], Linyun Yang [1], Yuxin Zhong[1], Xiang Xi [2], Zhenxiao Zhu [1], Xiaoyuan Jiao[1], Qing-an Tu[1], Yan Meng [2] ✉, Bei Yan [3] ✉, Ce Shang [4] ✉ & Zhen Gao [1,5,6] ✉

Topological polaritons characterized by light-matter interactions have become a pivotal platform for exploring new topological phases of matter. Recent theoretical advances unveiled a novel mechanism for tuning the topological phases of polaritons by modifying the surrounding photonic environment (light-matter interactions) without altering the lattice structure. Here, by embedding a dimerized chain of microwave helical resonators within a metallic cavity, we report the experimental observation of tunable topological phases of polaritons by varying the cavity width, which governs the strength of light-matter interactions. Moreover, we experimentally verified a previously predicted new type of topological phase transition, including three noncoincident critical points in the parameter space: the closure of the polaritonic bandgap, the transition of the Zak phase, and the hybridization of the topological edge states with the bulk states. Our experimental results reveal some unobserved properties of topological phases of matter when strongly coupled to light and provide a new design principle for tunable topological photonic devices.

Recent advances in topological photonics[1-5] and topological polaritonics[6-14] have revolutionized our ability to manipulate light, transcending the conventional boundaries of photonics. These breakthroughs have not only fostered a deeper understanding of the light-matter interactions at a fundamental level but also opened entirely new avenues for various applications in diverse fields, such as topological waveguides[15,16], cavities[17,18], lasers[8,17,19-21], integrated photonic circuits[22,23], nonlinear[24-29], and non-Hermitian photonics[30-32]. However, it is notoriously difficult, if not impossible, to manipulate the topological phases without modifying their lattice structures, since their topological invariants, such as the Zak phase in the one-dimensional (1D)[33] Su-Schrieffer-Heeger (SSH) model[34] and the Chern number in the two-dimensional (2D) topological photonic systems[35,36], are intrinsically determined by their lattice configurations.

On the other hand, controlling light-matter interactions with cavities has played a fundamental role in modern science and technologies such as cavity quantum electrodynamics[37-41], cavity magnonics[42-44], and cavity plasmonics[45-47]. More interestingly, recent theoretical studies[48-55] revealed that cavity-tunable light-matter interactions can act as a new degree of freedom to tune the topological phases of polaritons by modulating the surrounding photonic environment without changing the lattice structure, which cannot be achieved in the conventional 1D SSH chain of polariton micropillars[8] or the cavity-embedded SSH chain of dielectric resonators[25]. This novel mechanism has led to the theoretical discoveries of many previously unexplored topological phenomena, such as the breakdown of the celebrated bulk-boundary correspondence[48], the manipulation of type-I and type-II Dirac polaritons[49], and the tunable pseudomagnetic

[1]Department of Electronic and Electrical Engineering, Southern University of Science and Technology, Shenzhen, China. [2]School of Electrical Engineering and Intelligentization, Dongguan University of Technology, Dongguan, China. [3]College of Science, Wuhan University of Science and Technology, Wuhan, China. [4]Aerospace Information Research Institute, Chinese Academy of Sciences, Beijing, China. [5]State Key Laboratory of Optical Fiber and Cable Manufacture Technology, Southern University of Science and Technology, Shenzhen, China. [6]Guangdong Key Laboratory of Integrated Optoelectronics Intellisense, Southern University of Science and Technology, Shenzhen, China. [7]These authors contributed equally: Dong Zhao, Ziyao Wang. ✉e-mail: mengyan@dgut.edu.cn; yanbei@wust.edu.cn; shangce@aircas.ac.cn; gaoz@sustech.edu.cn

fields[50]. However, experimental observation of the cavity-tunable topological phase of polaritons has not yet been reported.

Here, we present the first experimental demonstration of the cavity-tunable topological phase of polaritons following the theoretical prediction in ref. [48]. These exotic topological phases of polaritons arise from the strong coupling between the cavity modes (representing light) and the dipolar modes (representing matter) by embedding a 1D dimerized chain of microwave helical resonators (MHRs) in a metallic cavity, which is distinct from conventional plasmon polaritons propagating along the structured metal surface. When embedded in a metallic cavity, the MHRs exhibit dipole moments oriented along the helical axis, with electric fields tightly localized in the air gap between the MHRs and the metallic cavity. This configuration represents a microwave version of polaritons in a dimerized chain of plasmonic nanorods embedded in a metallic cavity[49]. We experimentally demonstrate that the intrinsic band topology (Zak phase) and the polaritonic band structure of the composite structure can be fundamentally tuned by changing the cavity width. Moreover, we experimentally identify three noncoincident critical points in the parameter space: when the polaritonic bandgap closes, when the Zak phase changes from nontrivial to trivial, and when the topological edge states begin to hybridize with the bulk states, verifying a new type of topological phase transition that includes three different critical transition points[48,56].

## Results

### Design and modeling of light-matter interactions

To elucidate the formation of polaritons, we first study the light-matter interactions of two dipolar MHRs embedded within a metallic cavity depicting a three-level system comprising two coupled dipolar modes and a photonic cavity mode. As illustrated in the upper panel of Fig. 1a, two coupled dipolar modes are indicated by two eigenmodes of a pair of MHRs with coupling strength $\Omega$: a symmetric mode with a higher eigenfrequency $\omega_+^{dp}$ and an antisymmetric mode with a lower eigenfrequency $\omega_-^{dp}$. The resonance spectrum of these coupled dipolar modes is depicted in the lower panel of Fig. 1a, revealing two pronounced resonance peaks. The photonic cavity mode is shown in the

upper panel of Fig. 1b, displaying a fundamental resonance peak near the eigenfrequency $\omega_0^{ph}$ (lower panel). The embedding of coupled MHRs within a cavity, as illustrated in Fig. 1c, results in collective strong light-matter interaction according to cavity quantum electrodynamics[37,38,57], generating hybrid polaritonic modes with resonance frequencies of $\omega_L^{pol}$, $\omega_U^{pol}$, and $\omega_P^{pol}$, respectively. In Fig. 1d, the mode hybridization in this three-level system is depicted with one photonic cavity mode, two dipolar MHRs modes, and three polaritonic modes, providing insight into the interplay between the MHRs and the metallic cavity. The schematic representation employs a color scheme to delineate the relative occupations of dipolar and cavity modes within the system. The red hues indicate the predominant impact of the dipolar modes, whereas the blue hues signify the impact of the cavity modes. Two of the polaritonic modes $\omega_L^{pol}$ and $\omega_U^{pol}$ exhibit lower eigenfrequencies stemming from their original dipolar states $\omega_\pm^{dp}$, while the third polaritonic mode $\omega_P^{pol}$, mainly influenced by cavity modes, displays a higher eigenfrequency than the original cavity mode[58].

To demonstrate the cavity-tunable topological phases of polaritons, we embed a dimerized chain of MHRs in a metallic cavity with two air gaps (1 mm) separating the MHRs from the upper and lower metallic plates (a foam spacer fills the air gaps), as schematically shown in Fig. 2a. The metallic cavity has a fixed height of $L_z = 24$ mm, while its width $L_y$ is tunable to modulate the surrounding photonic environment and light-matter-interaction strength. Each unit cell (white cubic frame) consists of two MHRs and has a lattice constant of $d = 40$ mm. The alternating center-to-center distances between two neighboring MHRs are $d_1 = 0.575d$ and $d_2 = 0.425d$, respectively. A single MHR is shown in Fig. 2b with copper wire diameter $2r = 2$ mm, helix diameter $2R = 15$ mm, helix height $h = 22$ mm, axial intercept $l = 5$ mm, and 4 turns. The 1D dimerized chain of MHRs supports collective dipolar (dp) excitations (oscillating electric dipoles)[59], which can be modeled as a prototypical 1D SSH model in the microwave regime:

$$H_{dp} = \begin{pmatrix} \omega_0 & t_1 + t_2 e^{-ik_x d} \\ t_1 + t_2 e^{ik_x d} & \omega_0 \end{pmatrix}, \quad (1)$$

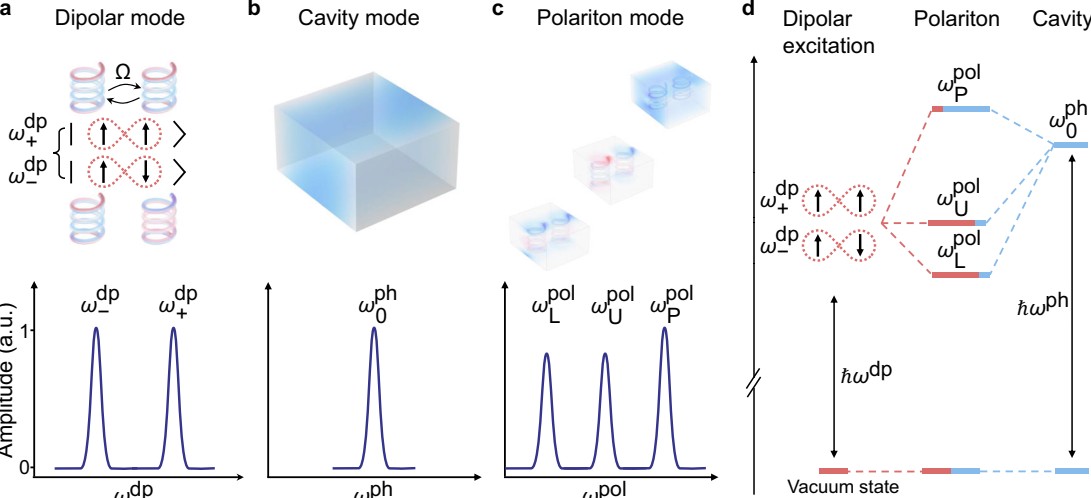

**Fig. 1 | Schematic illustration of light-matter interactions in a three-level system. a** The $E_z$ field distributions of the symmetric and antisymmetric eigenmodes of two coupled MHRs (upper panel). The Coulomb interaction ($\Omega$) results in mode splitting and two coupled dipolar modes with eigenfrequencies $\omega_+^{dp}$ (symmetric mode) and $\omega_-^{dp}$ (antisymmetric mode). These coupled dipolar modes display two resonance peaks near their respective eigenfrequency regimes (lower panel). a.u., arbitrary units. **b** The $E_z$ field distribution of a metallic cavity mode (upper panel) which exhibits a single resonance peak near its fundamental resonance frequency

$\omega_0^{ph}$ (lower panel). **c** The $E_z$ field distributions of the three polaritonic modes of two MHRs embedded in a metallic cavity (upper panel). The mode hybridization between the dipolar and cavity modes generates three polaritonic modes with eigenfrequencies $\omega_L^{pol}$, $\omega_U^{pol}$, and $\omega_P^{pol}$ (lower panel), respectively. **d** Mode hybridization of a three-level system comprising two coupled dipolar modes and a photonic cavity mode. The relative ratios of dipolar and cavity modes are qualitatively indicated by red and blue hues, respectively.

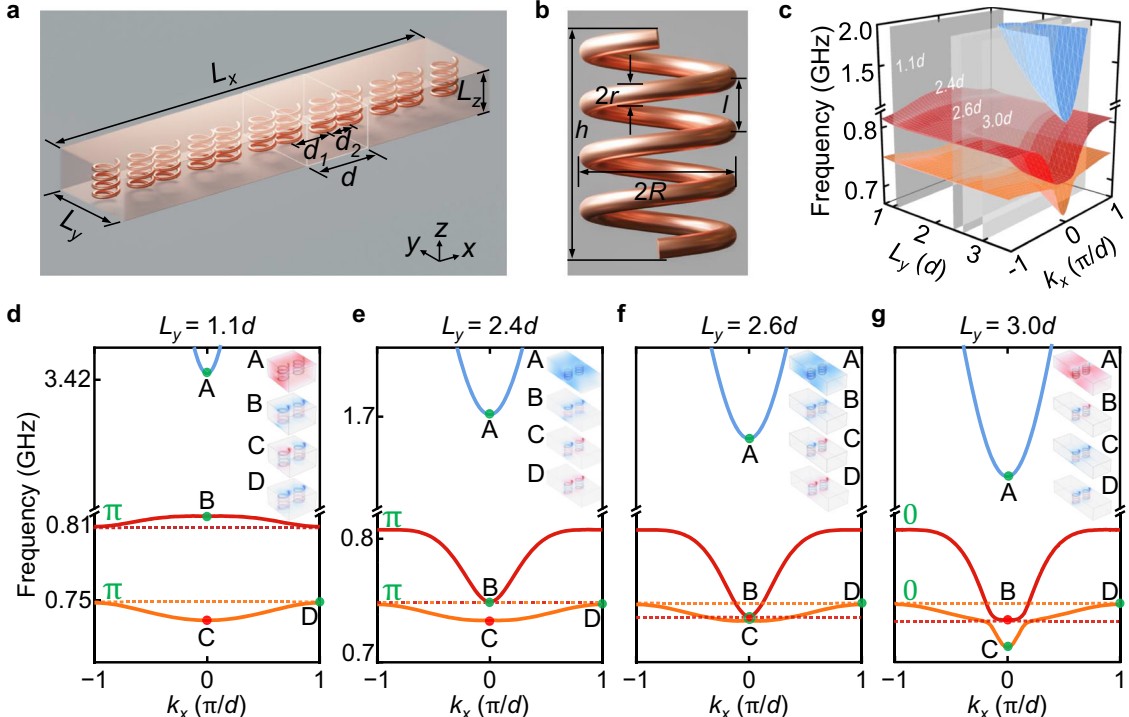

**Fig. 2 | Cavity-tunable topological phases of polaritons. a** Schematic of the composite structure consisting of a 1D dimerized chain of MHRs embedded in a metallic cavity. Comprising an MHR dimer, the chain's unit cell gives rise to two length scales: the alternating center-to-center distances $d_1$ and $d_2$. The lattice constant $d$ equals $d_1 + d_2$. The variables $L_x$, $L_y$, and $L_z$ denote the dimensions of the cavity along the x, y, and z directions, respectively. **b** A single MHR with a copper wire diameter of $2r$, a helix diameter of $2R$, a helix height of $h$, and a helix axial intercept of $l$. **c** Simulated band structure diagram of the composite structure as a function of cavity width $L_y$ and wavevector $k_x$. The three polaritonic bands are represented by the blue (higher), red (middle), and orange (lower) surfaces, respectively. Simulated band structures with different cavity widths $L_y = 1.1d$ (**d**), 2.4$d$ (**e**), 2.6$d$ (**f**) and 3.0$d$ (**g**), respectively. Green letters $\pi$ or 0 represent the Zak phases of the lowest two polaritonic bands ($\omega_L^{pol}$ and $\omega_U^{pol}$). Insets show $E_z$ field distributions of the eigenmodes at $k_x = 0$ (A, B, C) and $k_x = \pi/d$ (D). The red (yellow) dashed lines denote the bottom (top) of the lowest red (yellow) polaritonic band.

where $\omega_0$ is the resonance frequency, $t_1 = \Omega(d/d_1)^3$ [$t_2 = \Omega(d/d_2)^3$] is the intracell [intercell] coupling, $\Omega = \omega_0(a/d)^3/2$ is the coupling constant, $a$ is a length scale characterizing the strength of the dipolar excitations. The resulting band structure for the dipolar excitations is given as $\omega_\pm^{dp} = \omega_0 \pm \Omega|g_{k_x}|$, which shows a symmetric feature between the in-phase dipole momentum $\omega_+^{dp}$ and out-of-phase dipole momentum $\omega_-^{dp}$ with a bandgap of $2\Omega|g_{k_x}|$.

When embedded within a metallic cavity, the dipolar excitations will couple with the fundamental photonic (ph) cavity modes characterized by the dispersion relation of $\omega_{k_x}^{ph} = c_0\sqrt{k_x^2 + (\pi/L_y)^2}$, where $c_0$ represents the speed of light in vacuum (see Supplementary Information's Supplementary Note 1). This composite structure induces strong light-matter interactions between two dipolar modes and a photonic cavity mode, upgrading the typical SU(2) SSH model (1D dimerized chain of MHRs) to a SU(3) polaritonic model:

$$H_{pol} = \begin{pmatrix} \omega_0 & t_1 + t_2 e^{-ik_x d} & i\xi_{k_x} e^{-i\chi_{k_x}} \\ t_1 + t_2 e^{ik_x d} & \omega_0 & i\xi_{k_x} e^{i\chi_{k_x}} \\ -i\xi_{k_x} e^{i\chi_{k_x}} & -i\xi_{k_x} e^{-i\chi_{k_x}} & \omega_{k_x}^{ph} \end{pmatrix}, \quad (2)$$

where $\xi_{k_x} = \left(2\pi a^3 \omega_0^3 / dL_y L_z \omega_{k_x}^{ph}\right)^{1/2}$ indicates the strength of light-matter interactions, and $\chi_{k_x} = k_x d_1/2$ stems from the phase difference between two inequivalent lattice sites within a unit cell (see Supplementary Information's Supplementary Note 1 and 3)[48].

## Numerical simulation of cavity-tunable topological phases of polaritons

We use the COMSOL Multiphysics RF Module to solve the phase diagram of the polaritonic band structures $\omega_{j=\{L,U,P\}}^{pol}$ as a function of $L_y$ and $k_x$, as shown in Fig. 2c. For $L_y < 1.3d$, the polaritonic bands $\omega_{L,U}^{pol}$ (red and orange surface sheets) are smoothly deformed from the dipolar bands $\omega_\pm^{dp}$ due to the negligible weak light-matter interactions $\xi_{k_x}$; for $1.3d < L_y < 2.6d$, the center region (near $k_x = 0$) of $\omega_U^{pol}$ (red surface sheet) descends via the increase of $\xi_{k_x}$ while $\omega_L^{pol}$ (orange surface sheet) remains almost unchanged, and the polaritonic bandgap closes at $L_y = 2.4d$; for $L_y > 2.6d$, $\omega_{L,U}^{pol}$ display an anti-crossing phenomenon, pushing down the center region of $\omega_U^{pol}$ (orange surface sheet) while keeping $\omega_L^{pol}$ (red surface sheet) almost unchanged. We select four critical cavity widths ($L_y = 1.1d$, 2.4$d$, 2.6$d$, and 3.0$d$, respectively) and plot their simulated band structures in Fig. 2d–g with the $E_z$ field distributions of the eigenmodes at $k_x = 0$ (A, B, C) and $k_x = \pi/d$ (D) shown in the insets.

Notably, the change of surrounding photonic environments (light-matter-interaction strengths) not only modulates the band structures but also tunes the topological phases of polaritons (see Supplementary Information's Supplementary Note 2-1.1)[60]:

$$\theta_j^{Zak} = i\int_{-\pi/d}^{+\pi/d} dk_x \langle \psi_{k_x,j} | \partial_{k_x} | \psi_{k_x,j} \rangle, \quad (3)$$

which is quantized by 0 (topological trivial phase) or $\pi$ (topological nontrivial phase), where $\psi_{k_x,j}$ is the periodic part of the eigenstates. A

discretized form of Eq. (3) is defined by

$$\theta_j^{\mathrm{Zak}} = -\operatorname{Im}\sum_{n=1}^{N-1}\ln\left\langle\psi_{k_{x_n},j}|\psi_{k_{x_{n+1}},j}\right\rangle, \qquad (4)$$

where $N$ is the number of divided parts of $k_x$, $\psi_{j,k_{x_n}}$ is the discretized form of $\psi_{k_x,j}$ for a given momentum $k_{x_n}$. For the transverse magnetic modes corresponding to the electric field component $E_z$, $\psi_{k_{x_n},j}$ can be numerically extracted from the equation: $\psi_{k_{x_n},j} = E_{z;k_{x_n},j}e^{-ik_x x}$, where $E_{z;k_{x_n},j}$ represents the normalized $E_z$ in one unit cell. The Zak phases of $\omega_{\mathrm{L}}^{\mathrm{pol}}$ with different $L_y$ are presented as green letters in Fig. 2d–g, indicating the topological non-trivial (trivial) phases associated with $\pi$ (0) for $L_y < 2.6d \left(L_y > 2.6d\right)$. Additionally, the Zak phase can also be qualitatively determined by the symmetry of the eigenmodes at $k_x = 0$ (mode C) and $k_x = \pi/d$ (mode D). If they exhibit different (same) mode symmetries, the Zak phase is $\pi$ (0)[60,61].

## Experimental demonstration of cavity-tunable polaritonic band structures

To experimentally demonstrate the cavity-tunable topological phases of polaritons, we fabricate a sample comprising a dimerized chain of 30 MHRs embedded in a metallic cavity, as shown in Fig. 3a, where the upper metallic plate has been removed to see the inner structure. Each unit cell comprises two MHRs labeled site-1 and site-2, respectively, as shown in Fig. 3b. The experimental setup consists of a vector network analyzer (Agilent 5232A) and two electric monopole antennas, one as a point source to excite the composite structure and the other as a probe to measure the $E_z$ field distributions. We perform a Fast Fourier transform (FFT) of the $E_z$ field distributions to extract the measured polaritonic band structures (color maps), which agree well with the simulated results (white solid lines), as shown in Fig. 3c–f. These experimental results unambiguously verify that the polaritonic band structures can be modified by only structuring the surrounding photonic environment (light-matter interactions).

To experimentally explore the topological phase transition, we used 15 eigenmodes to calculate the Zak phase of the lowest two polaritonic bands. The Zak phases of the lowest two polaritonic bands are extracted with 15 eigenmodes [$N = 15$ in Eq. (4)], which experimentally reveal the topological phase transition as we tune the cavity width $L_y$. In the experiment, the eigenmodes $\psi_{k_x}$, which contain two elements $(F_{k_x}^O, F_{k_x}^E)$, are captured by employing a two-step measurement process. First, we measure the $E_z$ field distributions at all odd sublattices in each unit cell (i.e., sites 1, 3, 5, ..., 29) and apply FFT to the measured $E_z$ field distributions. For each wavevector $k_{x_n}$, the peak frequency in the FFT spectrum is regarded as the eigenfrequency, and the amplitude of the peak ($F_{k_{x_n}}^O$) is captured as the first component of the eigenmode $\psi_{k_{x_n}}$. Second, the same FFT is applied to the measured $E_z$ field distributions at all even sublattices (i.e., sites 2, 4, 6, ..., 30). The amplitude of the peak ($F_{k_{x_n}}^E$) in the FFT spectrum for the same wavevector $k_{x_n}$ is extracted as the second component of the eigenmodes. Consequently, the eigenmodes $\psi_{k_{x_n}}$ for any specific $k_{x_n}$ can be captured from the FFT spectra of all odd and even sublattices, i.e., $\psi_{k_x} = (F_{k_x}^O, F_{k_x}^E)$[62]. With the experimentally extracted $\psi_{k_{x_n},\mathrm{L}}$, the Zak phase of the $\omega_{\mathrm{L}}^{\mathrm{pol}}$ band can be obtained according to Eq. (4). As shown in Fig. 4a, the measured (red circles) and simulated (black line) Zak phases of the $\omega_{\mathrm{L}}^{\mathrm{pol}}$ band are plotted as a function of $L_y$. It can be observed that when we increase $L_y$ from 1.1$d$ to 3.5$d$, the Zak phases evolve from being nontrivial ($\pi$) to trivial (0) with a topological phase transition point around $L_y = 2.6d$.

## Experimental observation of cavity-tunable topological edge states

Finally, we investigate the manipulation of topological edge states by modulating the cavity width $L_y$. Fig. 4b shows the measured (color map) and simulated (white dots and cyan circles represent the bulk and edge states, respectively) polaritonic eigenfrequency spectra as a function of $L_y$. When $L_y$ is small, two bright lines within the bulk band gaps can be observed, which represent the measured topological edge states. As $L_y$

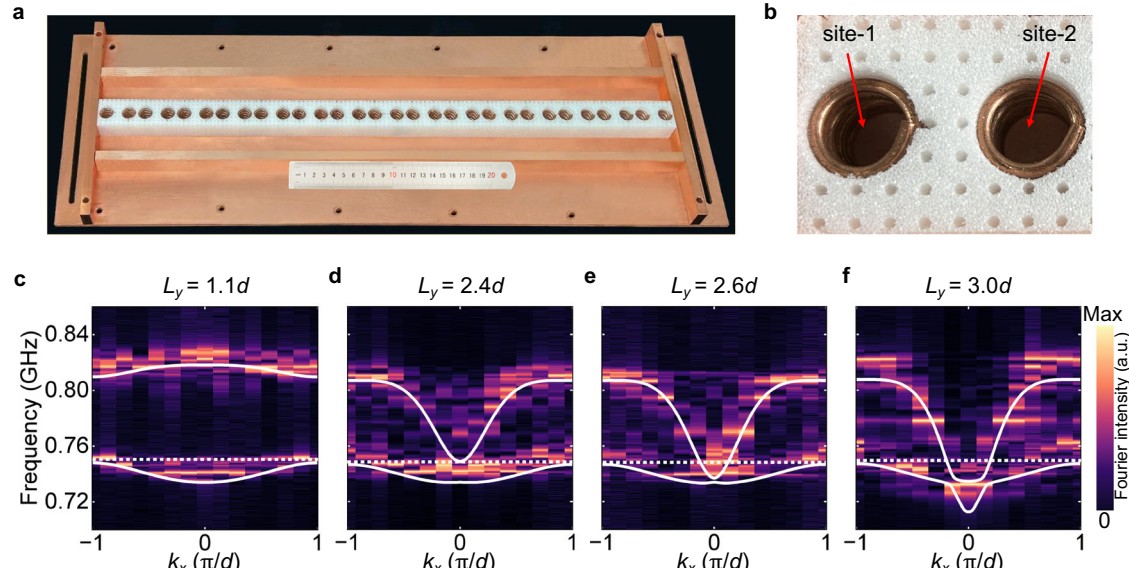

**Fig. 3 | Experimental observation of tunable polaritonic band structures by varying the cavity width $L_y$. a** Photo of the fabricated sample consisting of a 1D dimerized chain of 30 MHRs embedded in a metallic cavity. The upper metallic plate has been removed to see the inner structure. **b** A unit cell of the 1D dimerized chain of MHRs. The positions of the two MHRs in the unit cell are denoted as site-1 and site-2, respectively. **c–f** Measured (color maps) and simulated (white solid lines) polaritonic band structures of the sample with different cavity widths $L_y$. The white dashed lines represent the maximum frequency of the lowest polaritonic band ($\omega_{\mathrm{L}}^{\mathrm{pol}}$). a.u., arbitrary units.

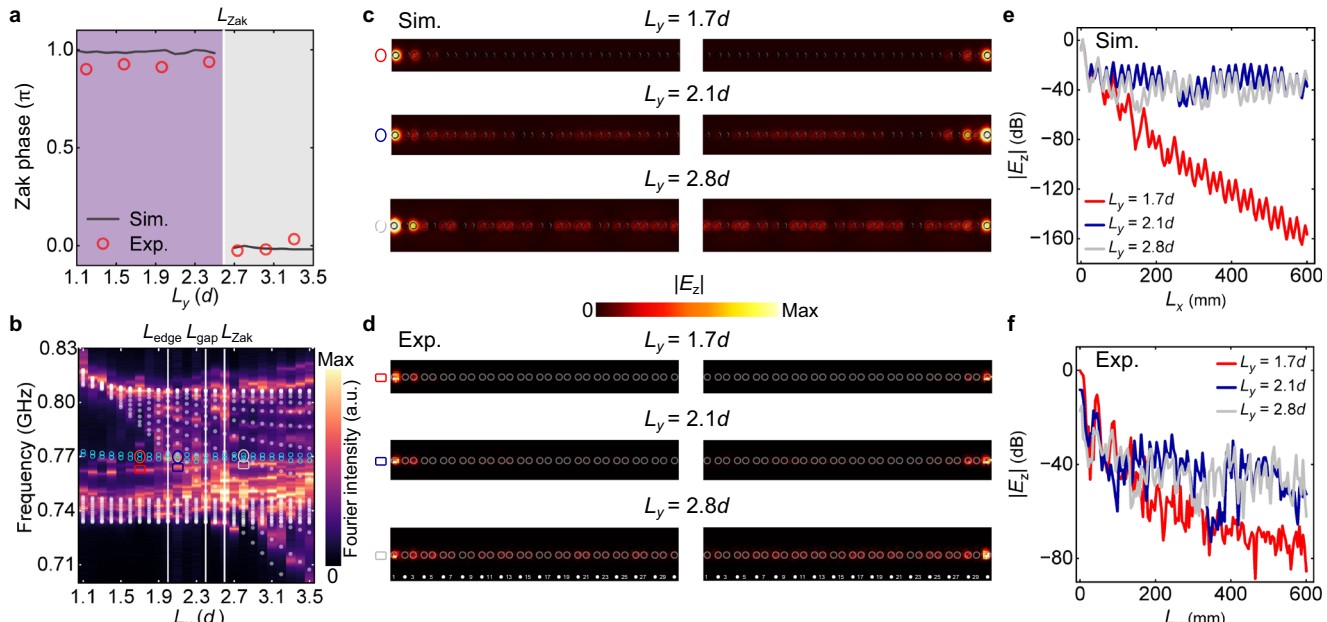

**Fig. 4 | Experimental observation of tunable topological phases of polaritons by changing the cavity width $L_y$. a** Measured (red circles) and simulated (black lines) Zak phases of $\omega_L^{pol}$ as a function of $L_y$. **b** Measured (color map) and simulated (white dots and cyan circles represent the bulk and edge states, respectively) eigenfrequency diagram of a 1D finite dimerized chain of 30 MHRs embedded in a metallic cavity as a function of $L_y$. a.u., arbitrary units. Simulated (**c**) and measured (**d**) electric field distributions of the topological edge states with $L_y = 1.7d$ (red ellipse and rectangle), 2.1$d$ (blue ellipse and rectangle), and 2.8$d$ (gray ellipse and rectangle), respectively. The gray circles and dots mark the locations of the MHRs. Simulated (**e**) and measured (**f**) $|E_z|$ field distributions (logarithmic representation) of the left topological edge states as a function of the cavity length $L_x$ with different cavity widths $L_y = 1.7d$ (red line), 2.1$d$ (blue line), and 2.8$d$ (gray line), respectively. Sim. (Exp.) denotes simulation (experiment).

increases from 1.1$d$ to 3.5$d$, the eigenfrequency spectra of the upper polaritonic band ($\omega_U^{pol}$) gradually approach those of the lower polaritonic band ($\omega_L^{pol}$), while the topological edge states remain almost unchanged. Consequently, the topological edge states gradually merge into the upper bulk band at $L_{edge} = 2.0d$ (left vertical white line), causing delocalization of the topological edge states to the bulk. Remarkably, this merging occurs before the polaritonic bandgap closes at $L_{gap} = 2.4d$ (middle vertical white line) and the Zak phase changes at $L_{Zak} = 2.6d$ (right vertical white line), which is totally different from the standard SSH model[8,17,25–27] or even its extended version that involves beyond the nearest-neighbor couplings[63]. Next, we show how the topological edge states evolve as we continuously increase $L_y$. Fig. 4c and d show the simulated and measured $E_z$ field distributions of the topological edge states for $L_y = 1.7d$ (red ellipse and rectangle), 2.1$d$ (blue ellipse and rectangle), and 2.8$d$ (gray ellipse and rectangle), respectively. We observe that before merging into the bulk band ($L_y < L_{edge}$), the topological edge states are located within the polaritonic bandgap and exhibit strong field localization at the left or right edge of the 1D MHRs chain (red ellipse and rectangle). However, when they merge into the bulk band ($L_y > L_{edge}$), the topological edge states begin to hybridize with the bulk states (blue ellipse and rectangle), resulting in novel mixed states (bound states in the continuum) with strong field localization near the edge and extended field distribution within the bulk simultaneously[48,56]. When we further increase $L_y$, the electric field distributions extend into the entire sample (gray ellipse and rectangle). Fig. 4e, f show the simulated and measured $|E_z|$ field distributions (logarithmic representation) of the left topological edge states as a function of the composite structure length $x$. It can be seen that for $L_y = 1.7d$ (red line), the topological edge states are tightly localized on the left boundary and decay exponentially along the composite structure. While for $L_y = 2.1d$ (blue line) and $L_y = 2.8d$ (cyan line), the topological edge states hybridize with the bulk states and distribute almost uniformly along the composite structure. The topological edge states at the right boundary exhibit similar behaviors (see Supplementary Information's Supplementary Note 3. and

Supplementary Fig. 7 for details). Note that the slight frequency discrepancy between the measured and simulated topological bulk and edge states originates from the fabrication error of the experimental sample.

## Discussion
In conclusion, we have experimentally demonstrated the cavity-tunable topological phases of polaritons for the first time, verifying the theoretical prediction in ref. 48. A novel three-level system is constructed by embedding a 1D dimerized chain of MHRs in a metallic cavity to elucidate the formation of topological polaritons. Notably, we have experimentally demonstrated that both the topological polaritonic band structures and topological invariant (Zak phase) can be tuned by modifying the surrounding photonic environment (light-matter interactions) without altering the lattice structure. Furthermore, we experimentally identified a new type of topological phase transition, which includes three noncoincident critical points in the parameter space: the closure of the polaritonic bandgap, the transition of the Zak phase, and the merging of the topological edge states with the bulk states. We envision that the proposed platform can be extended to 2D systems with more intriguing and abundant topological phenomena, such as cavity-tunable Dirac polaritons, valley edge states, and pseudo-magnetic fields. Our work not only demonstrates a novel mechanism for tuning topological phases but also provides a new mechanism for designing tunable topological photonic devices by employing strong light-matter interactions, which may have potential applications in tunable topological photonic devices[64,65], controllable light-matter interactions of polaritons[66–68], topological polariton lasing[13], and nonlinear topological polaritonic devices[69,70].

## Methods
### Numerical simulation
In this work, all numerical simulations are performed using the finite element method (FEM) with the radio frequency module of COMSOL

Multiphysics. The metallic cavity and microwave helical resonators are modeled as perfect electric conductors (PECs). The band structures and eigenfrequency spectra of the sample are obtained using the eigenfrequency solver. The band structures are computed by applying periodic boundary conditions along the $x$ direction, while all other surfaces are designated as PECs. The eigenfrequency spectra are calculated by solving for the eigenfrequencies of a sample consisting of 15 unit cells. Frequency domain calculations are employed to determine the electric field distributions of the tunable topological edge states.

## Sample fabrication

The experimental samples are fabricated using precision mechanical machining. The microwave helical resonators are constructed from T1-type copper wire (1 mm diameter), wound using a spring coiling machine (TY-S1.2). The copper plates constructing the metallic cavity are cut and perforated using a CNC lathe (Heidenhain iTNC 530). A perforated air foam (ROHACELL 31 HF with a relative permittivity of 1.04 and a loss tangent of 0.0025) is used to fix the microwave helical resonators.

## Experiment setup and measurement

In the microwave experiment, we use a vector network analyzer (VNA) to generate and detect microwave signals. Two electric dipole antennas serving as the source and probe are connected to the VNA to excite and measure the $E_z$ fields (both amplitude and phase) of the bulk and edge states of the experimental sample. The spatial electric field distributions are obtained by scanning the sample point by point using a stepper motor. By performing a Fast Fourier Transform (FFT) to the measured $E_z$ field distributions, the measured band structures can be obtained.

## Data availability

The data in the manuscript and supplementary information can be obtained from the Figshare database at https://doi.org/10.6084/m9. figshare.29086502.

## Code availability

We use the commercial software COMSOL Multiphysics to conduct electromagnetic numerical simulations. For requests regarding computation details, please contact the corresponding authors; the codes can be accessed from the Figshare database at https://doi.org/10. 6084/m9.figshare.29086502.

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

## Acknowledgements

Z.G. acknowledges the funding from the National Natural Science Foundation of China under grants No. 62361166627, 62375118, and 12104211, Guangdong Basic and Applied Basic Research Foundation under grant No.2024A1515012770, Shenzhen Science and Technology Innovation Commission under grants No. 20220815111105001 and 202308073000209, High level of special funds under grant No. G03034K004. Y. M acknowledges the support from the National Natural Science Foundation of China under Grant No. 12304484, and the Guangdong Basic and Applied Basic Research Foundation under grant No. 2024A1515011371.

## Author contributions

Z.G. conceived the idea and initiated the project; Z.G., D.Z., Y.M. and B.Y. designed the sample and built the experimental setup; D.Z., Z.Y.W. and B.Y. conducted the experimental measurements; D.Z., Z.G., B.Y., Y.M. and L.Y.Y. analyzed the data; Z.G., D.Z., B.Y., C.S. and Y.M. prepared the main manuscript; Y.X.Z., X.X., Z.X.Z., X.Y.J. and Q.A.T. participated in discussions of this project; Z.G., Y.M. and C.S. supervised the project.

## Competing interests

The authors declare no competing interests.
