## [Transparent Peer Review file · Nature Communications]

Observation of cavity-tunable topological phases of polaritons

Corresponding Author: Professor Yan Meng

Version 0:

Reviewer comments:

Reviewer #1

(Remarks to the Author)

In this rather impressive experimental work of Zhao and co-workers they present (to my knowledge) the first experimental realization of an increasingly popular model in quantum optics (that of a matter interacting with a multimode cavity, rather than the celebrated single photonic mode models of the famous Rabi, Dicke, Jaynes-Cummings, Tavis-Cummings models). They have designed and made a robust structure hosting resonators which mimic very well a topological multi-mode cavity quantum optics model. This is an impressive achievement, which will be welcomed by many international groups working on multimode cavities and quantum topology (see for example some papers [Communications Physics volume 5, Article number: 271 (2022)], [Phys. Rev. B 109, 155160 – Published 25 April 2024], [<https://arxiv.org/abs/2407.12088>], [Phys. Chem. Chem. Phys., 2022, 24, 15860-15870]), therefore I expect the work to have a large impact.

In their nice experiments, the authors probe the rich topological phase diagram of a SSH-like model in a multimode cavity, and they find experimentally a novel topological transition with implications for our understanding of the bulk-edge correspondence, as well as for the applicability of the Zak phase topological index in systems with strong light-matter coupling. These are important results for the expanding field of topological nanophotonics.

Overall, this high-quality research seems to be technically correct, and the topic is very fashionable and perfect for the journal. The presentation of the paper is also quite high, and the writing is at an accessible level making the results readable for a large audience. However, a few minor issues could be improved to make the paper even better (as suggested in my report) before publication.

Some comments:

1) The authors have very nicely shown a platform for exploring cavity-controlled lattice topology in 1-D. Could a similar setup be adapted for 2D lattices for example? What other experimental challenges would this dimensionality present? Could it be mentioned as another avenue of research opened up by this work in the concluding paragraphs, especially due to the cavity 2D theory works of Refs. 49 and 50 - where perhaps topological corner and hinge states could be similarly observed?

2) The reference list can be updated with some relevant and recent theory works, see for example [Communications Physics volume 5, Article number: 271 (2022)], [Phys. Rev. B 109, 155160 – Published 25 April 2024], [<https://arxiv.org/abs/2407.12088>], [Phys. Chem. Chem. Phys., 2022, 24, 15860-15870]. This experimental work will certainly be welcomed by many theoretical groups working worldwide on multimode cavity models very similar to the one realized neatly here, which goes beyond celebrated single mode cavity models (e.g. Tavis-Cummings, Rabi, Dicke, Jaynes-Cummings models).

3) 30 microwave helical resonators are used in the experiment – why was this number chosen? Is this an experimental limitation or is this the smallest number of resonators which corresponds to the results of a continuum theory? Are there any size-effects present due to this choice? Presumably, an even number of resonators needs to be chosen?

4) Can the code used to generate the numerical simulations data be made freely available to meet the scientific ideals of openness and reproducibility, perhaps on github or on the authors website?

Reviewer #2

(Remarks to the Author)

The manuscript presents an experimental demonstration of a predicted breakdown of the celebrated bulk-edge correspondence in a system comprising an SSH chain strongly coupled to a cavity mode.

The physical realization comprises MHR resonators embedded in a metallic cavity, working in the GHz regime. The manuscript also includes numerical COMSOL simulations and a simplified model (given in the supplementary material) that captures the essential physics.

The findings are very interesting, and experimental confirmation of the predictions are both timely and necessary.

Nevertheless, before publication, the experimental results must be clarified and set correctly in light of the predictions.

Concerning the experimental results:

- Panel 4b is critical, as it provides extensive experimental information.

However, the current format makes it challenging to interpret.

The cyan dots, representing simulations, mask possible experimental results in a region that is critical for interpretation. It is unclear from the present figure if there is a substantial Fourier intensity at the (L_y , frequency) values where simulated modes appear.

The plot shows an experimental mode at the smallest L_y values at a frequency slightly above 0.76 GHz, while the prediction shows two modes at approximately 0.77 GHz. However, the comparison of the spatial profile of the modes (panels 4c and d) is @0.77 GHz.

Have the authors considered only one of the simulated modes for this comparison? In that case, was it chosen because it is the one that fits best? What about the other simulated mode? What about the experimental mode at @0.76 GHz?

It would also be helpful to show the spatial profiles of the predicted and measured modes in a simple x-y plot on a logarithmic scale to visualize the exponential localization of the modes, along with an estimate of the experimental error.

Concerning the presentation:

- The manuscript provides experimental confirmation of a previously predicted phenomenon (ref. 48 in the manuscript). This is a great result. But still, it is a confirmation. The abstract currently lacks mention of this prior prediction. Instead, it states that "we experimentally identified a new type of topological phase transition..." and "the results reveal some remarkable and uncharted properties...". In my opinion, the abstract should indicate that this work experimentally confirms a previously predicted new topological phase transition.

Similarly, the supplemental material must be presented correctly, given the similarities with the supplementary material (SM) in their Ref. 48 (Ref 5. in the SM). A clearer introduction, such as "The main text builds on a model first proposed theoretically in Ref. [5], with further details provided here for completeness," would help clarify this connection. Adding a statement like "Here, we provide a detailed account of the model, closely following Ref. [5]" would further clarify the relationship to prior work.

Towards the end of the SM, they say "the above section is derived from article [5]" but I think it should be clearly stated at the top of the SM not to be misleading to the journal readership.

In summary, this manuscript presents a necessary experimental confirmation of a predicted phenomenon. This is a significant achievement, but to recommend this manuscript for publication, the experimental results should be presented with greater clarity, and the findings should be contextualized appropriately within the existing theoretical framework.

Version 1:

Reviewer comments:

Reviewer #1

(Remarks to the Author)

The authors have satisfactorily addressed all of the comments raised by the reviewers, leading to a better and more readable manuscript. I am happy to support publication in its current form, I just have one final comment:

1) In the Supplemental Material it is said "Following the theoretical prediction in Ref. [1], here, we provide a detailed derivation of the model for completeness." The theory in this submitted SM is close to the theory presented in the SM of Ref. [1] - is it too close in the Editor's judgement or does there need to be a sentence, something like "We follow closely the presentation of Ref. [1]"?

I leave this to the Editor to decide.

Reviewer #2

(Remarks to the Author)

As mentioned in my previous report, the manuscript presents the experimental confirmation of an intriguing prediction: the failure of the bulk-edge correspondence when a topological system is placed inside a cavity. The findings are very interesting, and experimental confirmation of the predictions is both timely and necessary.

The authors have satisfactorily addressed my two comments. I believe the results are now presented in a clearer way, and the authors' contributions are placed in proper context.

I thus recommend the publication of the manuscript in Nature Communications.

Response Letter to Referees

We are grateful for the constructive comments on our manuscript (NCOMMS-24-62766) from two Referees.

In the text below, Referee comments are quoted in *italics* and are followed by the corresponding detailed response. We also revised the Manuscript and the Supplementary Material based on the suggestions from both Referees, and these updates are highlighted in blue in those files. In the text below, the references to these updates are highlighted in a similar way.

GENERAL COMMENTS FROM REFEREE #1:

In this rather impressive experimental work of Zhao and co-workers they present (to my knowledge) the first experimental realization of an increasingly popular model in quantum optics (that of a matter interacting with a multimode cavity, rather than the celebrated single photonic mode models of the famous Rabi, Dicke, Jaynes-Cummings, Tavis-Cummings models). They have designed and made a robust structure hosting resonators which mimic very well a topological multi-mode cavity quantum optics model. This is an impressive achievement, which will be welcomed by many international groups working on multimode cavities and quantum topology (see for example some papers [Communications Physics volume 5, Article number: 271 (2022)], [Phys. Rev. B 109, 155160 – Published 25 April 2024], [<https://arxiv.org/abs/2407.12088>], [Phys. Chem. Chem. Phys., 2022, 24, 15860-15870]), therefore I expect the work to have a large impact.

In their nice experiments, the authors probe the rich topological phase diagram of a SSH-like model in a multimode cavity, and they find experimentally a novel topological transition with implications for our understanding of the bulk-edge correspondence, as well as for the applicability of the Zak phase topological index in systems with strong light-matter coupling. These are important results for the expanding field of topological nanophotonics.

Overall, this high-quality research seems to be technically correct, and the topic is very fashionable and perfect for the journal. The presentation of the paper is also quite high, and the writing is at an accessible level making the results readable for a large audience. However, a few minor issues could be improved to make the paper even better (as suggested in my report) before publication.

Response from Authors:

We thank Referee #1 for his/her appreciation that our work is a “*rather impressive experimental work*”, “*the first experimental realization of an increasingly popular model in quantum optics*”, and “*the topic is very fashionable and perfect for the journal*”. We have thoroughly revised our Manuscript and Supplementary Material based on his/her suggestions and comments by incorporating additional experimental and numerical results. In the following, we will address Referee#1’s specific comments

point-by-point.

SPECIFIC COMMENTS FROM REFEREE #1:

Referee #1 -- Comment 1:

The authors have very nicely shown a platform for exploring cavity-controlled lattice topology in 1-D. Could a similar setup be adapted for 2D lattices for example? What other experimental challenges would this dimensionality present? Could it be mentioned as another avenue of research opened up by this work in the concluding paragraphs, especially due to the cavity 2D theory works of Refs. 49 and 50 - where perhaps topological corner and hinge states could be similarly observed?

Response from Authors:

We thank Referee #1 for raising these interesting and inspiring questions. Yes, a similar setup to be adapted for 2D lattices is possible, and in fact, following Ref. 49 (Nat. Commun. 9, 2194 (2018)), we have conducted another experimental work to manipulate the Dirac polaritons by embedding a 2D honeycomb metasurface in a metallic cavity. Compared with the 1D light-matter interaction system, which can be tuned by changing L_y or L_z , the main experimental challenge of the 2D system is we can only change L_z to tune the light-matter interaction, as adopted in Refs. 49 and 50. Thus, we need to redesign the microwave helical resonator with a large metallic cap to increase the interaction between the microwave helical resonators and the parallel metallic plates (cavity). In a 2D light-matter interaction system, we think the topological corner states can be realized and tuned by modifying the light-matter interaction. However, the topological hinge state can only be realized in a 3D system, it seems that we have no degree of freedom to introduce the light-matter interaction to tune the topological hinge state. We have included a brief discussion of this potential research direction on Page 6 in the revised manuscript: “We envision that the proposed platform can be extended to 2D systems with more intriguing and abundant topological phenomena, such as the cavity-tunable Dirac polaritons, valley edge states, and pseudo-magnetic fields.”

Referee #1 -- Comment 2:

The reference list can be updated with some relevant and recent theory works, see for example [Communications Physics volume 5, Article number: 271 (2022)], [Phys. Rev. B 109, 155160 – Published 25 April 2024], [<https://arxiv.org/abs/2407.12088>], [Phys. Chem. Chem. Phys., 2022, 24, 15860-15870]. This experimental work will certainly be welcomed by many theoretical groups working worldwide on multimode cavity models very similar to the one realized neatly here, which goes beyond celebrated single mode cavity models (e.g. Tavis-Cummings, Rabi, Dicke, Jaynes-Cummings models).

Response from Authors:

We thank Referee #1 for his/her constructive suggestion to update the relevant and recent theoretical works. These works indeed complement our experimental findings and will enhance the context and relevance of our research. We have added these references in the revised manuscript, including [Commun. Phys. 5, 271 (2022) as Ref. 53; Phys. Rev. B 109, 155160 (2024) as Ref. 55; Phys. Chem. Chem. Phys. 24, 15860

(2022) as Ref. 52; and arXiv:2407.12088 (2024) as Ref. 54].

Referee #1 -- Comment 3:

30 microwave helical resonators are used in the experiment – why was this number chosen? Is this an experimental limitation or is this the smallest number of resonators which corresponds to the results of a continuum theory? Are there any size-effects present due to this choice? Presumably, an even number of resonators needs to be chosen?

Response from Authors:

We thank Referee #1 for his/her questions on the number of microwave helical resonators we used in the experiment. In the original theoretical study [Phys. Rev. Lett. **123**, 217401 (2019)], 500 dimers of nanorods were considered, corresponding to 1000 microwave helical resonators in our system. In principle, the more microwave helical resonators, the more consistent the experimental results with the theoretical model. However, considering fabrication constraints and experimental limitations, we utilized 15 unit cells (30 microwave helical resonators) in the experiment, which is sufficient to resolve the band structure and demonstrate the tunability of topological edge states. Note that the finite-size effect induces a slight eigenfrequency splitting in both the simulated and experimental results in Fig. 4(b) in the manuscript. To maintain an integer number of unit cells (dimers) requires selecting an even number of resonators.

To clarify this point, we have conducted additional simulations of the 1D composite structures consisting of 20, 30, and 40 microwave helical resonators, as shown in Fig. R1. We can see that similar tunability of the topological edge states can be obtained for 20, 30, and 40 microwave helical resonators. In our manuscript, 30 microwave helical resonators were chosen to balance the accuracy and the experimental feasibility.

Fig. R1 | Comparison of the eigenfrequency spectra and the topological edge states for configurations with 20 (a-c), 30 (d-f), and 40 (g-i) microwave helical resonators, respectively.

Referee #1 -- Comment 4:

Can the code used to generate the numerical simulations data be made freely available to meet the scientific ideals of openness and reproducibility, perhaps on github or on the authors website?

Response from Authors:

We thank Referee #1 for his/her constructive suggestion. We fully agree with the importance of reproducibility and openness in scientific research. We will make the code used to generate the numerical simulation data freely available in a public GitHub repository named identically to the article's formal title. Moreover, we will also provide the code if the reader needs it. This can ensure that other researchers can reproduce our results.

GENERAL COMMENTS FROM REFEREE #2:

The manuscript presents an experimental demonstration of a predicted breakdown of the celebrated bulk-edge correspondence in a system comprising an SSH chain strongly coupled to a cavity mode.

The physical realization comprises MHR resonators embedded in a metallic cavity, working in the GHz regime. The manuscript also includes numerical COMSOL simulations and a simplified model (given in the supplementary material) that captures the essential physics.

The findings are very interesting, and experimental confirmation of the predictions are both timely and necessary.

In summary, this manuscript presents a necessary experimental confirmation of a predicted phenomenon. This is a significant achievement, but to recommend this manuscript for publication, the experimental results should be presented with greater clarity, and the findings should be contextualized appropriately within the existing theoretical framework.

Response from Authors:

We thank Referee #2 for his/her appreciation that “*The findings are very interesting, and experimental confirmation of the predictions are both timely and necessary*”. In response to Referee #2’s comments, we have thoroughly revised our manuscript with additional numerical and experimental results. In the following, we will address Referee #2’s specific comments point-by-point.

Referee #2 -- Comment 1:

a) Panel 4b is critical, as it provides extensive experimental information. However, the current format makes it challenging to interpret. The cyan dots, representing simulations, mask possible experimental results in a region that is critical for interpretation. It is unclear from the present figure if there is a substantial Fourier intensity at the (L_y , frequency) values where simulated modes appear.

Response from Authors:

We thank Referee #2 for his/her constructive comments. To resolve the issue of experimental results being obscured by the simulated results, we have replaced the cyan dots with cyan circles in the revised manuscript. Now the substantial Fourier intensity can be clearly observed near 0.77 GHz, as shown in the revised Fig. 4b.

b) The plot shows an experimental mode at the smallest L_y values at a frequency slightly above 0.76 GHz, while the prediction shows two modes at approximately 0.77 GHz. However, the comparison of the spatial profile of the modes (panels 4c and d) is @0.77 GHz. Have the authors considered only one of the simulated modes for this comparison? In that case, was it chosen because it is the one that fits best? What about the other

simulated mode? What about the experimental mode at @0.76 GHz?

Response from Authors:

We are sorry for the misinterpretation caused by the presentation in Fig. 4b. In fact, there are two topological edge states in the measured Fourier intensity of the experimental results, and the topological edge states with a higher frequency are covered by the simulation results. In the original manuscript, we only compared the simulated and measured topological edge states localized at the left boundary (higher frequency), while the simulated topological edge states localized at the right boundary (lower frequency) are presented in Fig. S7 of the Supplementary Materials. In the revised manuscript, we remeasured the topological edge states localized at the right boundary (0.76 GHz) and presented the measured and simulated topological edge states at both the left and right boundaries in Figs. 4c-4d. Moreover, we also show the $|E_z|$ field distribution (logarithmic representation) of the left (right) topological edge states as a function of the composite structure length x with different cavity widths in Figs. 4e-4f (Fig. S7). These new results enhance the clarity and strengthen the experimental observations of topological edge states.

c) It would also be helpful to show the spatial profiles of the predicted and measured modes in a simple x-y plot on a logarithmic scale to visualize the exponential localization of the modes, along with an estimate of the experimental error.

Response from Authors:

We thank Referee #2 for his/her constructive suggestion. In the revised manuscript, we have added the simulated and measured $|E_z|$ field distribution (logarithmic representation) of the left topological edge states as a function of cavity length x in Figs. 4e–4f, as shown in Fig. R2. For the right topological edge states, their corresponding simulated and measured $|E_z|$ field distribution (logarithmic representation) as a function of cavity length x is presented in Supplementary Material Fig. S7.

Fig. R1 | Experimental observation of tunable topological phases of polaritons by changing the cavity width L_y . a Measured (red circle) and simulated (black lines) Zak phases

of ω_L^{pol} as a function of L_y . **b** Measured (color map) and simulated (white dots and cyan circles represent the bulk and edge states, respectively) eigenfrequency diagram of a 1D finite dimerized chain of 30 MHRs embedded in a metallic cavity as a function of L_y . **c-d** Simulated (c) and measured (d) electric field distributions of the topological edge states with $L_y = 1.7d$ (red ellipse and rectangle), $2.1d$ (blue ellipse and rectangle), and $2.8d$ (gray ellipse and rectangle), respectively. The gray circles mark the locations of the MHRs. **e-f** Simulated (e) and measured (f) $|E_z|$ field distribution (logarithmic representation) of the left topological edge states as a function of the composite structure length x with different cavity widths $L_y = 1.7d$ (red line), $2.1d$ (blue line), and $2.8d$ (gray line), respectively.

To make this point clear, we have added the following discussion in the revised manuscript on Page 5: “Figs. 4e and 4f show the simulated and measured $|E_z|$ field distributions (logarithmic representation) of the left topological edge states as a function of the composite structure length x . It can be seen that for $L_y = 1.7d$ (red line), the topological edge states are tightly localized on the left boundary and decay exponentially along the composite structure. While for $L_y = 2.1d$ (blue line) and $L_y = 2.8d$ (cyan line), the topological edge states hybridize with the bulk states and distribute almost uniformly along the composite structure.”.

Referee #2 -- Comment 2:

The manuscript provides experimental confirmation of a previously predicted phenomenon (ref. 48 in the manuscript). This is a great result. But still, it is a confirmation. The abstract currently lacks mention of this prior prediction. Instead, it states that “we experimentally identified a new type of topological phase transition...” and “the results reveal some remarkable and uncharted properties...”. In my opinion, the abstract should indicate that this work experimentally confirms a previously predicted new topological phase transition.

Similarly, the supplemental material must be presented correctly, given the similarities with the supplementary material (SM) in their Ref. 48 (Ref 5. in the SM). A clearer introduction, such as “The main text builds on a model first proposed theoretically in Ref. [5], with further details provided here for completeness,” would help clarify this connection. Adding a statement like “Here, we provide a detailed account of the model, closely following Ref. [5]” would further clarify the relationship to prior work. Towards the end of the SM, they say “the above section is derived from article [5]” but I think it should be clearly stated at the top of the SM not to be misleading to the journal readership.

Response from Authors:

We thank Referee #2 for his/her constructive suggestion and the attitude of seeking truth from the facts. We fully agree that our experimental work follows the theoretical prediction in Ref. 48. To highlight this point more clearly, we have revised the abstract, introduction, discussion, and Supplementary Material to explicitly state that our work

is an experimental verification of the previous theoretical prediction in Ref. 48. Specifically, we have modified the relevant sentences to: “Moreover, we experimentally verified a previously predicted new type of topological phase transition ...” and “Our experimental results reveal some unobserved properties of topological phases of matter when strongly coupled to light and provide a new design principle for tunable topological photonic devices.”

In the introduction, we added the following sentence: “Here, we present the first experimental demonstration of the cavity-tunable topological phase of polaritons following the theoretical prediction in [48].”

In the discussion, we changed the first sentence to: “In conclusion, we have experimentally demonstrated the cavity-tunable topological phases of polaritons for the first time, verifying the theoretical prediction in [48].”

In the Supplementary Material, we have highlighted the connection between the Supplementary Material with the theoretical prediction in Ref. 48 (Ref. 1 in the SM). We have added the sentence: “Following the theoretical prediction in Ref. [1], here, we provide a detailed derivation of the model for completeness.”

We hope these statements prevent any potential confusion and explicitly state the relationship between our experimental work with the prior theoretical work.

Response Letter to Referees

We are grateful for the constructive comments on our manuscript (NCOMMS-24-62766A) from two Referees.

In the text below, Referee comments are quoted in *italics* and are followed by the corresponding detailed response. We also revised the Manuscript and the Supplementary Material based on the suggestions from Referees, and these updates are highlighted in blue in those files. In the text below, the references to these updates are highlighted in a similar way.

GENERAL COMMENTS FROM REFEREE #1:

The authors have satisfactorially addressed all of the comments raised by the reviewers, leading to a better and more readable manuscript. I am happy to support publication in its current form.

Response from Authors:

We thank Referee #1 for his/her recognition of our revised manuscript because the revisions have enhanced the manuscript's clarity and readability

SPECIFIC COMMENTS FROM REFEREE #1:

Referee #1 -- Comment 1:

I just have one final comment:

1) In the Supplemental Material it is said "Following the theoretical prediction in Ref. [1], here, we provide a detailed derivation of the model for completeness." The theory in this submitted SM is close to the theory presented in the SM of Ref. [1] - is it too close in the Editor's judgement or does there need to be a sentence, something like "We follow closely the presentation of Ref. [1]"??

Response from Authors:

We thank Referee #1 for his/her constructive suggestion. Based on Referee #1 suggestion, we have revised the first sentence of the Supplementary Material as follows:

Original text (Supplementary Material, Page 2):

“Following the theoretical prediction in Ref. [1], here, we provide a detailed derivation of the model for completeness.”

Revised text:

“For completeness, we follow closely the presentation of Ref. [1], providing details of the model.”

GENERAL COMMENTS FROM REFEREE #2:

As mentioned in my previous report, the manuscript presents the experimental confirmation of an intriguing prediction: the failure of the bulk-edge correspondence when a topological system is placed inside a cavity. The findings are very interesting, and experimental confirmation of the predictions is both timely and necessary.

The authors have satisfactorily addressed my two comments. I believe the results are now presented in a clearer way, and the authors' contributions are placed in proper context.

I thus recommend the publication of the manuscript in Nature Communications.

Response from Authors:

We thank Referee #2 for his/her insightful evaluation and constructive feedback on our manuscript. We are delighted that you find our experimental confirmation of the bulk-edge correspondence failure in cavity-embedded topological systems both timely and impactful.